# Study of the Two-Line Element Accuracy by 1U CubeSat with a GPS Receiver

**DOI:** 10.3390/s22082902

**Published:** 2022-04-10

**Authors:** Pavel Kovář, Pavel Puričer, Kateřina Kovářová

**Affiliations:** 1Department of Radioelectronics, Faculty of Electrical Engineering, The Czech Technical University in Prague, 166 27 Prague, Czech Republic; pavel.puricer@fel.cvut.cz; 2Department of Economics and Management, Jan Evangelista Purkyne University in Usti nad Labem, 400 96 Usti nad Labem, Czech Republic; kovarovak@g.ujep.cz; 3Department of Management, College of Hospitality Management and Economics in Prague, 181 00 Prague, Czech Republic

**Keywords:** LEO satellite orbit, two-line element set, simplified perturbations model

## Abstract

There is a common practice to calculate orbital trajectories of space objects like satellites and space debris using Two-Line Element Sets (TLEs). However, TLEs provide rather coarse parameters for fine orbit computation and their precision varies with age of their issue and position of the satellite. The paper evaluates such induced position determination error using the comparison of a position calculated from TLE data for a small CubeSat class satellite and a position obtained from the on-board custom GPS receiver that is a part of such satellite payload. The analyses of the impact of satellite position at the orbit, i.e., a dependency of position error on satellite geographical latitude, and impact of the ageing of TLE data in frame of position and velocity vector were made. There was shown that use of TLE data can bring some significant errors in calculation of predicted satellite position which can affect performance and efficiency of some related tasks like steering the ground station antenna for communication with the satellite or planning the satellites operations namely for the classes of small and amateur satellites.

## 1. Introduction

The NASA/NORAD Two-Line Element Set (TLE) is a de facto standard for the orbital elements description of a near-earth orbiting spacecraft (or space debris). The parameters were originally used for objects with orbital periods less than 225 min [1], and later extended to medium Earth orbits (MEO) and geostationary orbits (GEO).

The TLEs then become a usual data source for the algorithm of the satellite state vector calculations using simplified perturbations models (SGP, SGP4, SDP4, SGP8, and SDP8) [2]. TLE data are measured by a Space Surveillance Network (SSN) [3,4] and have been provided by the US government since 1970.

Since the satellites’ TLE is publicly available, e.g., in [5], and a number of professional and semi-professional software packages [6] or source codes exist for Matlab, C, FORTRAN, etc. [7,8], the TLE elements are widely used for many purposes, e.g., steering setup of the antennas for communication with small satellites, space experiment planning, and for processing and interpretation of scientific data measured by small satellites.

The precision of the position determination using TLE has been investigated by many researchers. The GOCE LEO satellite case was investigated in [9] on its orbit below 300 km, where most pieces of space debris are located. A position error lower than 2.5 km for the initial epoch time was observed; however, due to the observed rapid time degradation of the TLE precision, the error increased to 100 km in the frame of one week.

The results of similar research with Flock 1, Flock 1B, and Flock 1D CubeSat constellation of 3U are reported in [10], and another investigation of TLE precision for the Iridium constellation is mentioned in [11]. The orbit of Iridium satellites can be better predicted because of their height of approximately 780 km. Reference [12] analyzes the precision of TLE for the determination of the position of the GPS satellites. These satellites are placed on MEO orbit, which is nearly ideal for long-term orbit prediction, so the observed position errors are several times lower in comparison with the lower orbits for the same epoch time. Similar research [13] is focused on the study of the precision of time position determination of geostationary satellites. The estimation of the orbital parameters based on the GPS measurements is solved in [14,15].

The aim of this paper is to analyze the TLE position and velocity vectors’ prediction precision of the 1U unstabilized CubeSat class satellite. The position and velocity vectors calculated from TLE are compared with the position measured and velocity vectors measured by the on-board GPS receiver that is capable to reach precision up to tens of meters. As the satellite is not equipped with the attitude control system, the satellite slowly randomly rotates, and thus, its motion is practically uncontrolled in a same manner as the motion of space debris.

## 2. Materials and Methods

### 2.1. Lucky 7 CubeSat

Lucky 7 CubeSat (Figure 1) is a private CubeSat developed and launched by two enthusiasts Jaroslav Laifr and Pavel Kovar [16,17]. The satellite operates on a nearly circular polar orbit (eccentricity 0.0025 and inclination 97.6°) at a height of 520 km. The orbital period is approximately 93.5 min. The satellite (Figure 2) payload integrates an on-board computer block (OBC), scientific block module, and power supply block. The OBC consists of two (primary and backup) computers, UHF radio modems, and special watchdog that monitors the proper function of the computer and switches to the backup in a case of a failure of primary computer.

The scientific module is populated with a piNAV GPS receiver, a dosimeter, and a spectrometer for measurement of the space radiation. The last instrument is a 640 × 480 pixel resolution camera for taking snapshots from space.

The power supply subsystem consists of the 5 Gallium Arsenide solar panels with a peak power of about 2 W, power supply, and LiFe accumulators with a capacity of 4 Ah.

Radio modems operate in an amateur radio UHF band. The bit rate of the communication system link is 4.8 Kbits/s, which enables it to download up to 100 KBytes of data during one optimal satellite pass. The duration of the communication window is up to 10 min for the ground station located in the Czech Republic.

The memory of the on-board computer can store up to 2000 1-min samples of the data from all scientific instruments including GPS. The on-board computer further enables saving up to 4000 s of the GPS data with sampling frequency 1 s for detailed analyses of the GPS receiver behavior. The data records are transmitted from the satellite to the ground in a volume-saving format. The redundant data or data that can be calculated from other sources like azimuths and elevations of the navigation satellite are not recorded. The useful data are transformed to the binary format and properly quantized to reduce their binary representation. For example, the GPS positions are rounded to meters and GPS velocity to meters per second.

### 2.2. piNAV GPS Receiver

The GPS receiver piNAV (Figure 3) was developed purposefully for small satellite navigation at LEO orbits. The receiver is designed as a software receiver, where the digital signal and data processing runs in FPGA and microcontroller (Figure 4). The Lucky-7 satellite is equipped with a second version of the receiver that is augmented with the signal acquisition unit that considerably shortens the cold start to the receiver, i.e., the time to first position fix after switching on the receiver [17].

The main advantage of the receiver is the capability to provide not only standard NMEA message as do most GPS receivers, but also additional information, i.e., position and velocity vectors in ECEF coordinates as well as detailed receiver status.

The performance of the receiver was evaluated using a software GNSS simulator for various static and dynamic scenarios, namely the International Space Station and equatorial and polar orbits [16]. For the purpose of this paper, we can consider that the absolute position error of the GPS receiver is better than 10 m and absolute velocity is much smaller than the velocity quantization, which is 1 m/s. The time of measurement is quantized to hundreds of microseconds, which assures that the absolute time error is significantly smaller than quantization noise. The maximal satellite movement for the quantization step is less than 1 m.

### 2.3. Data Processing

The measured data were processed in Scilab software for numerical computing and data processing, namely CNES CelestLab and CelectLabX toolboxes [8]. These toolboxes integrate the Simplified Perturbation model and many supporting functions for the processing of the TLE and calculation of the satellite position.

The processing software compared position rGPS=(xGPS,yGPS,zGPS) and velocity vGPS=(vxGPS,vyGPS,vzGPS) measured by the GPS with the position rTLE=(xTLE,yTLE,zTLE) and velocity vTLE=(vxTLE,vyTLE,vzTLE) vectors calculated from TLE. All calculations were expressed in Earth-Centered Earth-Fixed (ECEF) coordinates. The errors of the GPS receivers were neglected as they are much lower than the expected errors of the TLE positions and velocity.

## 3. Results and Discussion

The measurement campaign ran in summer 2021, as seen in Table 1. There were four collected sets (TLE1–TLE4) of the TLE data, as described in Table 2. The epoch of the first set was 36 days before the experiment, while the last set was generated during the experiment. As the GPS position was not available 100% of the time due to the satellite rotation, we excluded such measurements with no GPS reference from the processing. We also excluded GPS measurements with poor geometry of the navigation satellites, i.e., with a high value of Position Dilution Of Precision (PDOP > 4).

Figure 5 shows the time dependency of the position error magnitude, i.e., a magnitude of the difference of GPS position and TLE-based position vectors. An expected degradation of the position precision with the increasing age of the TLE is visible.

Figure 6 displays the dependency of the position error magnitude on the satellite latitude. The position error is the highest near the equator and it decreases with an increase in latitude.

The position error vectors primarily calculated in ECEF frame were transformed to the Hill frame (Figure 7, Figure 8 and Figure 9), also known as the local-vertical, local-horizontal LVLH frame (Figure 10). The Hill frame is a local frame with the $x_{h}$ coordinate perpendicular to the reference ellipsoid, the $y_{h}$ coordinate laying in orbital plane, and the $z_{h}$ coordinate perpendicular to the $x_{h}y_{h}$ plane forming a right-handed orthogonal system. The transformation from the ECEF to the Hill frame can be done in three steps. (1) Transformation to the ENU (east, north, up) coordinates. (2) Rearrangement of the order of the coordinates. (3) Rotation around $x_{h}$ axis of the course over ground. The rotation angle can be calculated from the velocity vector, for instance.

The radius error |rGPS|−|rTLE| is drawn in Figure 11 and Figure 12. We can again observe an error increase with the age of the TLE data. The radius error dependency on the latitude, which creates closed cyclic curve that has area increasing with the age of ephemeris, is very interesting. Likely the errors in North–South part of the orbit are different to the errors in the other part.

It can be seen from Figure 10 that the error $y_{h}$ along the orbit depends on time. The error is almost independent of the position of the satellite in orbit, in contrast to the heights $x_{h}$ and cross errors $z_{h}$, which are orbit dependent.

The velocity error magnitude dependences on time and latitude are shown in Figure 13 and Figure 14. The velocity error is increasing with the age of the TLE elements. In contrast to the position error, the velocity error is the largest in the high latitudes and the lowest in equatorial one.

In Figure 15, Figure 16 and Figure 17, there are x,y, and z coordinates of velocity errors. The x and y coordinates errors show a positive correlation with the increasing TLE age in the equatorial regions. The increase in error around the poles is significantly smaller. The observed z coordinate velocity error shows latitude-dependent systematic errors that even swapped the sign over during aging.

The velocity error for TLE1 seems too high. From Figure 15, Figure 16 and Figure 17, it is evident the velocity vector alternates with the latitude. The sign of the *z* component is different in the South and North hemispheres. In addition, the sign of this velocity component between TLE1 and TLE2 was exchanged. This could be explained by the unmodeled influence of the Moon gravitation. The Moon orbital period is 28 days and the time distance between TLE1 and TLE2 is 12 days, which is nearly the half of the Moon orbital period.

## 4. Conclusions

The paper analyzed the calculations precision of the position and velocity vectors calculated from TLE for a small 1U unstabilized CubeSat in the polar orbit and the impact of the age of TLE to the precision. The novelty of the research can be seen in the analyses of the precision not only for the satellite position vector, but also for the velocity vector.

The next contribution of the paper was the investigation of the dependency of position precision on the satellite’s geographical latitude. This allowed us to state that the error of the position vector for the evaluated case has the largest values in the equatorial regions, while the error of the velocity is the smallest in same regions and vice versa.

The obtained precision of the position vector is comparable with other studies [9,10,18,19]. The precision of the velocity vector cannot be compared as these data were not available from previous studies.

The results can be quite well applied in the planning and processing of the satellite experiments and for the control of the ground antenna operations for communication with the satellites in cases where TLE data are the sole source for satellite position and velocity estimates.

## Figures and Tables

**Figure 1 sensors-22-02902-f001:**
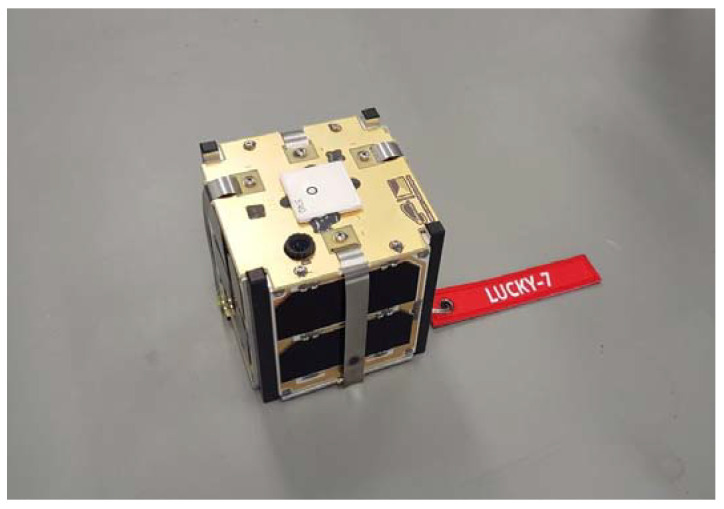
Lucky-7 CubeSat.

**Figure 2 sensors-22-02902-f002:**
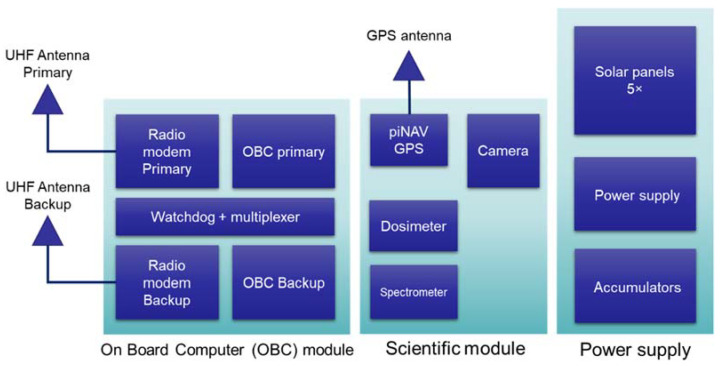
Block scheme of Lucky 7.

**Figure 3 sensors-22-02902-f003:**
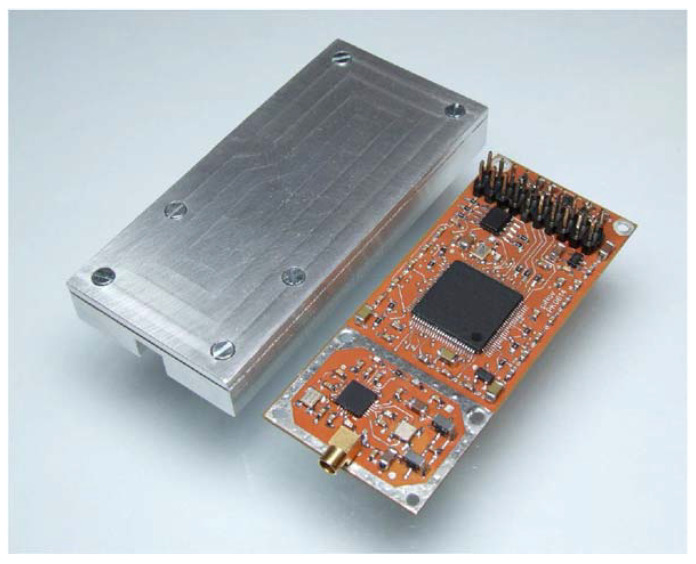
piNAV GPS receiver.

**Figure 4 sensors-22-02902-f004:**
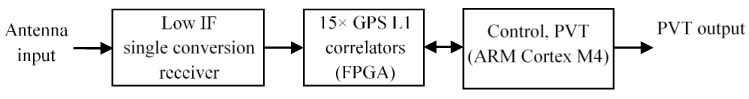
piNAV block scheme.

**Figure 5 sensors-22-02902-f005:**
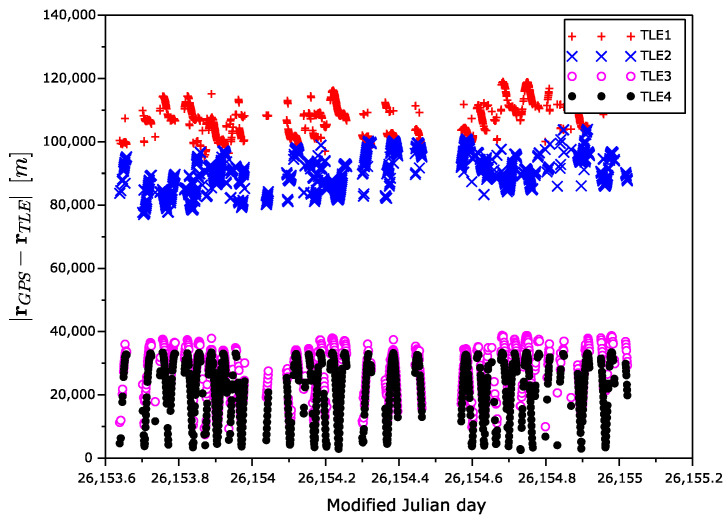
Position error magnitude as a function of time.

**Figure 6 sensors-22-02902-f006:**
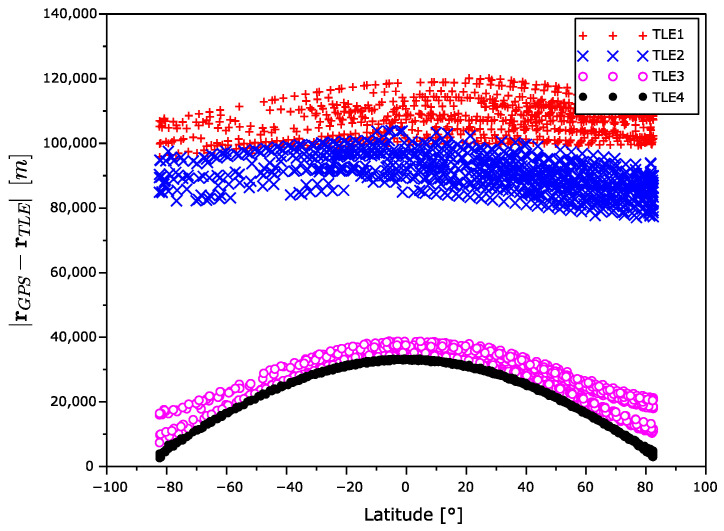
Position error magnitude as a function of latitude.

**Figure 7 sensors-22-02902-f007:**
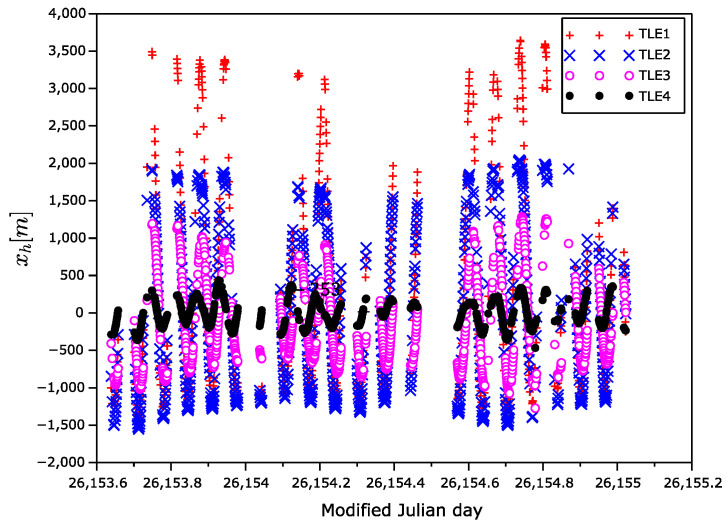
$x_{h}$ position error as a function of time in Hill frame.

**Figure 8 sensors-22-02902-f008:**
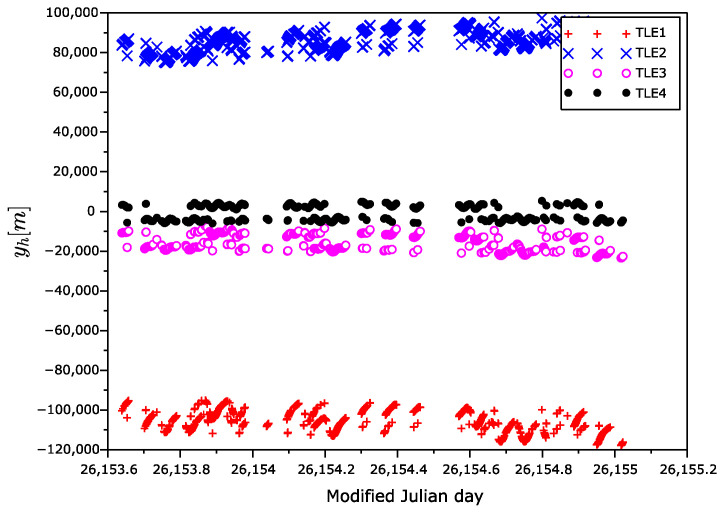
$y_{h}$ position error as a function of time in Hill frame.

**Figure 9 sensors-22-02902-f009:**
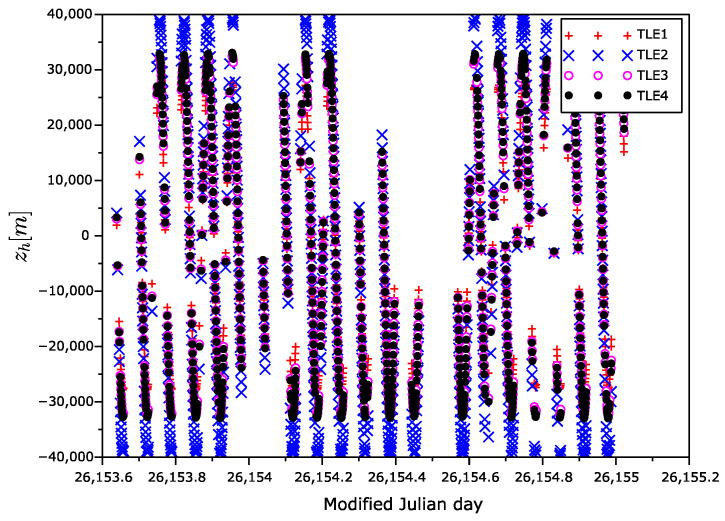
$z_{h}$ position error as a function of time in Hill frame.

**Figure 10 sensors-22-02902-f010:**
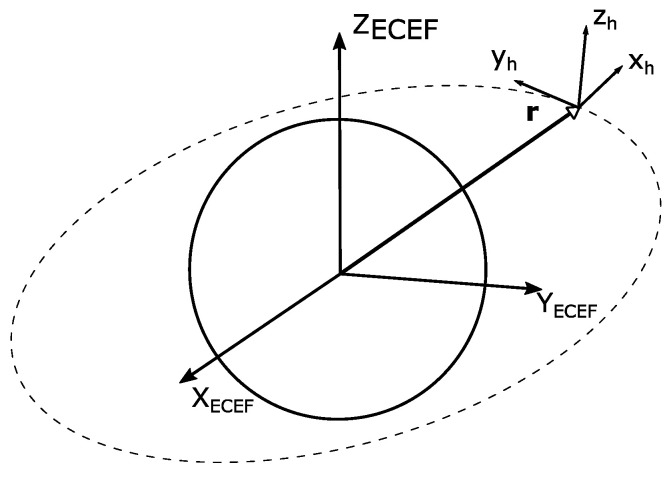
Hill frame and its relationship to ECEF.

**Figure 11 sensors-22-02902-f011:**
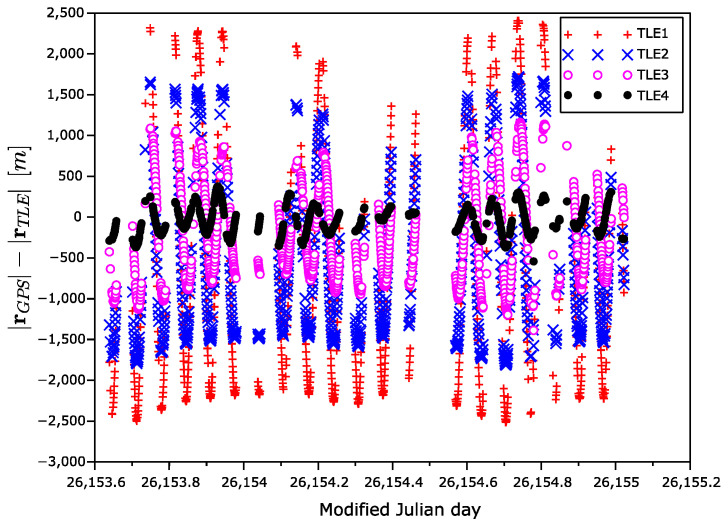
Radius error as a function of time.

**Figure 12 sensors-22-02902-f012:**
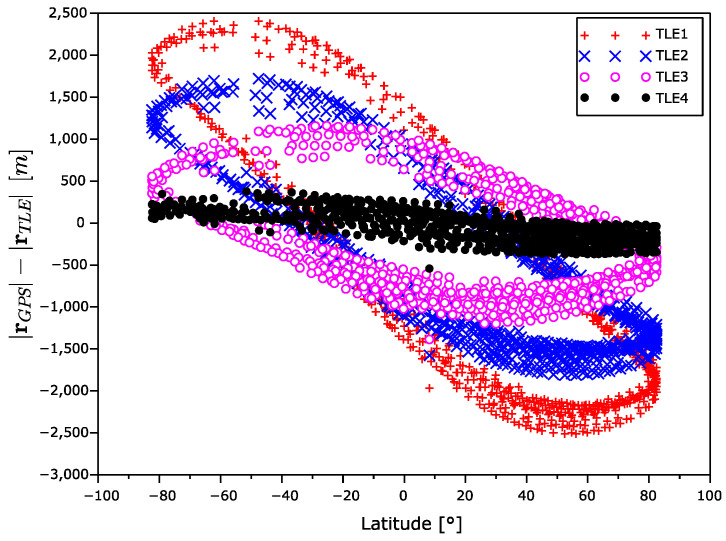
Radius error as a function of latitude.

**Figure 13 sensors-22-02902-f013:**
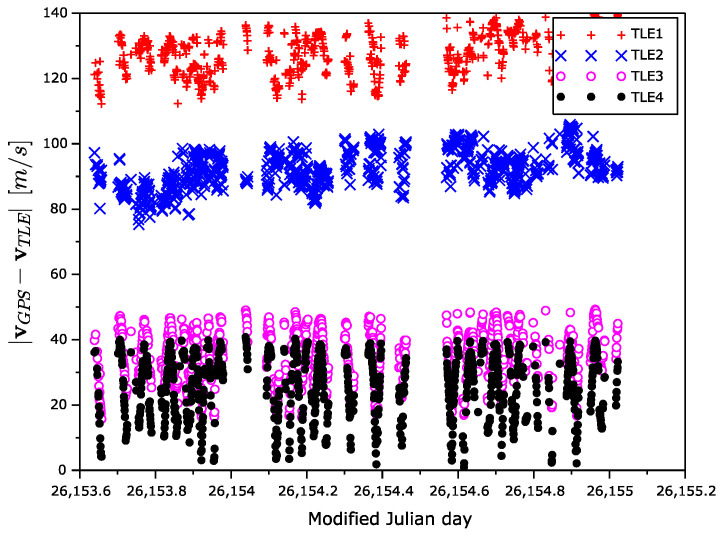
Velocity error magnitude as a function of time.

**Figure 14 sensors-22-02902-f014:**
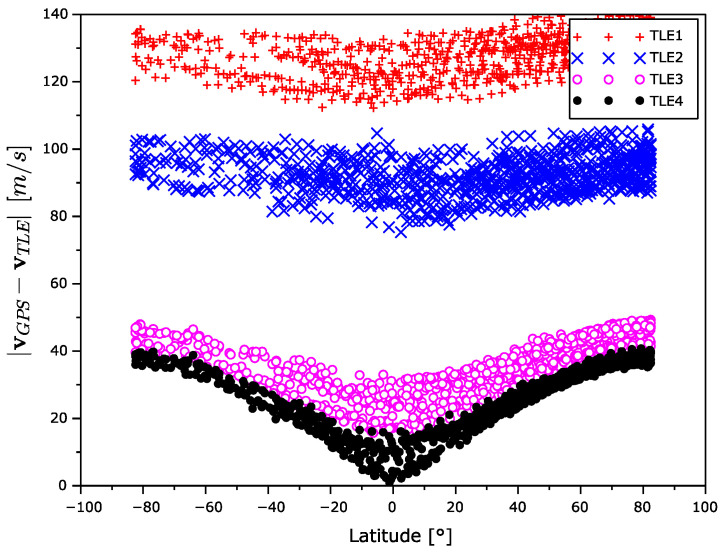
Velocity error magnitude as a function of latitude.

**Figure 15 sensors-22-02902-f015:**
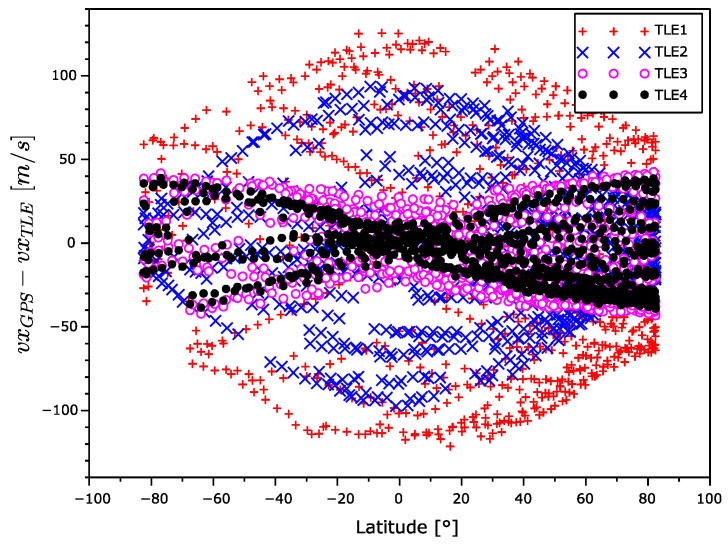
*x* coordinate velocity error as a function of latitude.

**Figure 16 sensors-22-02902-f016:**
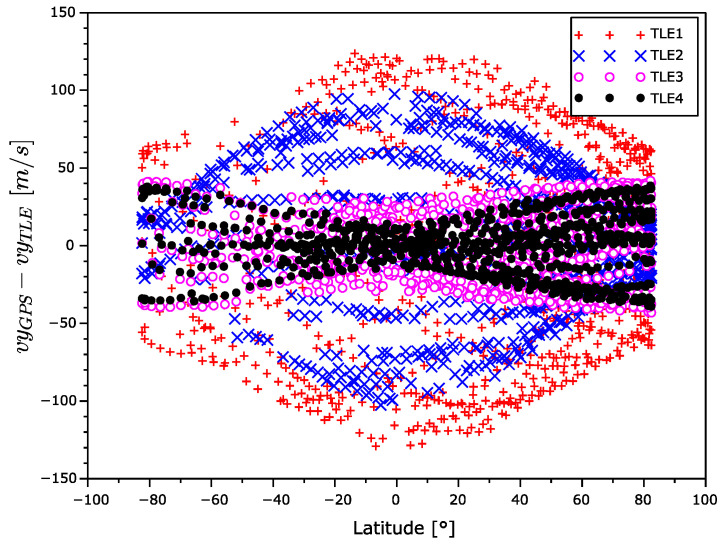
*y* coordinate velocity error as a function of latitude.

**Figure 17 sensors-22-02902-f017:**
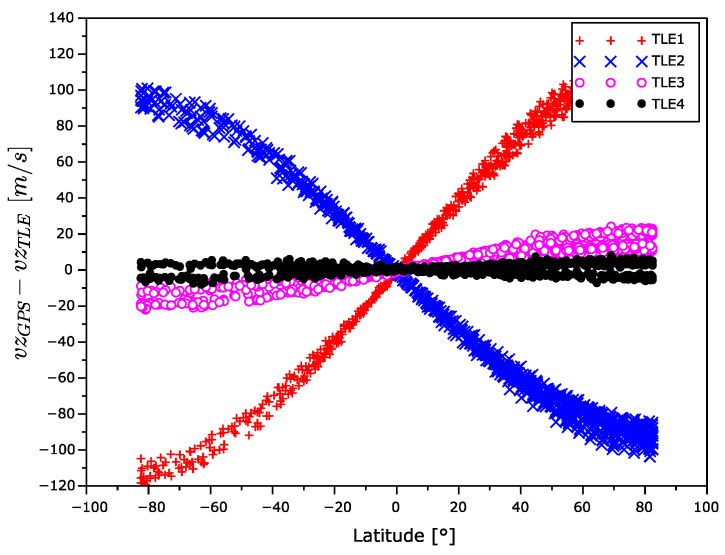
*z* coordinate velocity error as a function of latitude.

**Table 1 sensors-22-02902-t001:** Experiment time.

Experiment	Time (Modified Julian Day)	Time UTC
beginning	26,153.638952	9 August 2021 15:20:05.484
end	26,155.02235	11 August 2021 00:32:11.053

**Table 2 sensors-22-02902-t002:** TLE epoch.

TLE Designation	Epoch (Modified Julian Day)	Epoch (UTC)	Relationship between TLE Epoch and Experiment
TLE1	26,118.83682	5 July 2021 20:05:01.904	36 days before experiment
TLE2	26,130.87348	17 July 2021 20:57:48.978	24 days before experiment
TLE3	26,142.843845	29 July 2021 20:15:08.265	12 days before experiment
TLE4	26,154.880130	10 August 2021 21:07:23.280	During experiment

## Data Availability

The data that support the findings of this study are available from the author upon reasonable request.

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
