# Peer review of "Study of the Two-Line Element Accuracy by 1U CubeSat with a GPS Receiver"

_sensors, 2022, doi:10.3390/s22082902_

Round 1

Reviewer 1 Report

In this paper, a GPS receiver is installed on CubeSat satellite for position measurement, and the positions calculated by TLE data from different epoch are compared with the position measured by GPS receiver. The experimental results are analyzed from latitude and time, and the conclusion is reasonable. I suggest to enrich theoretical derivation process and supplement  calculation process of rGPS, vGPS, rTLE and vTLE in Section 2.3.

Reviewer 2 Report

This paper described the performance of  Two-Line Element Sets (TLE) based on small CubeSat class satellite. The results are interesting and the paper is well organized. Just two typos in the paper:

  1. In line 19, " There were made analyses " seems wired.
  2. In line 51, "nad" should be "and".

Reviewer 3 Report

The paper is devoted to the problem of TLE precision. No concern that the update rate of TLE is much slower than onboard GPS measurements. However authors describe approach that compare 36, 24 and 12 days old TLE to real-time GPS measurements. Why should we care about these old TLE while up-to-date ones are available? What is the TLE update rate for the satellite under investigation?

Given that approach we see that 12-days old TLE perform nearly the same as during-experiment TLE. That give as clue that TLE rather stable source of data. Lucky 7 satellite is described as having «polar orbit» but no other details are provided. Is it heliosynchronous orbit or geosynchronous polar orbit? The details must be important for the analysis since strong daily variations of plasma density in the ionosphere and the plasmasphere could be significant source of GPS positioning error. It becomes critical point when we take look on the receiver. While «receiver is capable to reach precision up to tens of the meters» the reference [17] says it is an L1-GPS receiver with C/A only measurements. The same is in datasheet https://www.skyfoxlabs.com/pdf/piNAV-NG_Datasheet_rev_F.pdf. Moreover the reference [17] explicitly says that «no ionospheric model is used». Plasmasphere is not mentioned at all. No field tests were performed only modeling. Single-frequency measurements does not allow construct ionosphere and geometry free combinations. So called ionosphere free combinations allow to reduce the error introduced by plasma on the path GPS satellite — GPS receiver. Ground based single-frequency receiver uses signals from GNSS base station to improve positioning. In space there are no base stations so precision is questionable.

One can see the sine wave pattern of the error envelope on Figure 10 as well as the sine wave pattern of the back dots curve. The envelope has a period of one day and the amplitude of black dots curve seems to be modulated with the same period of 1 day. Also there are different errors for the same TLE (say black dots) on the Figures 7, 8, 9. This says that GPS positioning is unstable, giving different results for different times of the day (sun zenith angle).

The wording should be improved for some point including (but not limited to):

Line 51: “nad” typo?

Line 52: “is better predictable”

Line 73: “correct function of the computer”

VGA resolution — VGA is the interface that supports a list of resolutions, please be more explicit in the definition of resolution, provide numbers.

«The maximal satellite movement for the quantization step is less than 1 m» - I am not sure that I understand the point here.

x(yz)-coordinate velocity → x component of the velocity vector?

Line 150: what coordinate system is used to define x,y,z?

Why does TLE3 and TLE4 seem to be PI shifted (better agreement on low latitudes) compared to TLE1 and TLE2 on the Figures 7 and 8 while on the Figure 9 the pattern is the same for all of them?

Given 4.8 Kbit / s and 100 Kbyte data volume is it correct that the signal from the satellite is received less than 180 s.

Is «4000 s of GPS data» enough to cover one loop? What is the orbital period?

While the design of the experiment could be acceptable. The data for the receiver that does not account for the error introduced by the plasma does not allow interpret results to make conclusions on TLE performance. And once again it is well known that old TLE does not perform well enough that is why they routinely update.

Reviewer 4 Report

See attached file.

Round 2

Reviewer 4 Report

The revised version generally includes the corrections listed in the review report.

The old versions of Figures 8-10 still appear in the paper, without titles. Do not forget to remove the figures.

Regarding the velocity error (lines 217-222): The authors state that the large velocity error is due to the effect of the Moon gravity. However, this effect is included in the SGP4 model that is used to predict the orbit from the TLE data, so I am not sure about this explanation.

Typos:

Line 33: The plural of spacecraft is spacecraft.

Line 135: Centered (not Cantered).

Author Response

Dear reviewer,

We appreciate for Reviewer’s warm work earnestly and hope that the corrections will meet with approval.

The old versions of Figures 8-10 still appear in the paper, without titles. Do not forget to remove the figures.

This problem is probably caused by an improper function of Word revisions. The problem was solved by the removal of these figures from the text.

Regarding the velocity error (lines 217-222): The authors state that the large velocity error is due to the effect of the Moon gravity. However, this effect is included in the SGP4 model that is used to predict the orbit from the TLE data, so I am not sure about this explanation.

Rigorous analysis of the problem is very complicated. We can only guess the cause. The problem was solved by the modification of the statement that the problem can be caused by an unmodeled influence of the Moon.

Typos:

Line 33: The plural of spacecraft is spacecraft.

Line 135: Centered (not Cantered).

Typos have been corrected.

Once again, thank you very much for your comments and suggestions.